# Sex Hormone-Binding Globulin (SHBG) in Cerebrospinal Fluid Does Not Discriminate between the Main FTLD Pathological Subtypes but Correlates with Cognitive Decline in FTLD Tauopathies

**DOI:** 10.3390/biom11101484

**Published:** 2021-10-08

**Authors:** Marta del Campo, Yolande A. L. Pijnenburg, Alice Chen-Plotkin, David J. Irwin, Murray Grossman, Harry A. M. Twaalfhoven, William T. Hu, Lieke H. Meeter, John van Swieten, Lisa Vermunt, Frans Martens, Annemieke C. Heijboer, Charlotte E. Teunissen

**Affiliations:** 1Neurochemistry Laboratory, Department of Clinical Chemistry, and Biobank, Amsterdam Neuroscience, Amsterdam University Medical Centers, Vrije Universiteit, 1081 HV Amsterdam, The Netherlands; ham.twaalfhoven@amsterdamumc.nl (H.A.M.T.); l.vermunt@amsterdamumc.nl (L.V.); c.teunissen@amsterdamumc.nl (C.E.T.); 2Departamento de Ciencias Farmacéuticas y de la Salud, Facultad de Farmacia, Universidad San Pablo-CEU, CEU Universities, 28668 Madrid, Spain; 3Department of Neurology, Alzheimer Center Amsterdam, Amsterdam Neuroscience, Amsterdam University Medical Centers, Location VUmc, 1081 HV Amsterdam, The Netherlands; yal.pijnenburg@amsterdamumc.nl; 4Department of Neurology, Perelman School of Medicine, University of Pennsylvania, Philadelphia, PA 19104, USA; chenplot@pennmedicine.upenn.edu (A.C.-P.); dirwin@pennmedicine.upenn.edu (D.J.I.); mgrossma@pennmedicine.upenn.edu (M.G.); 5Department of Neurology, Emory University School of Medicine, Atlanta, GA 30322, USA; william.hu@rutgers.edu; 6Rutgers-RWJ Medical School, Institute for Health, Health Care Policy, and Aging Research, Rutgers Biomedical and Health Sciences, New Brunswick, NJ 08901, USA; 7Department of Neurology and Alzheimer Center, Erasmus Medical Center Rotterdam, 3015 GD Rotterdam, The Netherlands; h.meeter@erasmusmc.nl (L.H.M.); j.c.vanswieten@erasmusmc.nl (J.v.S.); 8Endocrine Laboratory, Amsterdam University Medical Centers, Location AMC, 1105 AZ Amsterdam, The Netherlands; f.martens@amsterdamumc.nl (F.M.); a.heijboer@amsterdamumc.nl (A.C.H.)

**Keywords:** CSF, biomarkers, FTLD-Tau, FTLD-TDP

## Abstract

Biomarkers to discriminate the main pathologies underlying frontotemporal lobar degeneration (FTLD-Tau, FTLD-TDP) are lacking. Our previous FTLD cerebrospinal fluid (CSF) proteome study revealed that sex hormone-binding globulin (SHBG) was specifically increased in FTLD-Tau patients. Here we investigated the potential of CSF SHBG as a novel biomarker discriminating the main FTLD pathological subtypes. SHBG was measured in CSF samples from patients with FTLD-Tau (*n* = 23), FTLD-TDP (*n* = 29) and controls (*n* = 33) using an automated electro-chemiluminescent immunoassay. Differences in CSF SHBG levels across groups, as well as its association with CSF YKL40, pTau181/total-Tau ratio and cognitive function were analyzed. CSF SHBG did not differ across groups, though a trend towards elevated levels in FTLD-Tau cases compared to FTLD-TDP and controls was observed. CSF SHBG levels were not associated with either CSF YKL40 or the p/tTau ratio. They, however, inversely correlated with the MMSE score (r = −0.307, *p* = 0.011), an association likely driven by the FTLD-Tau group (r FTLD-Tau = −0.38; r FTLD-TDP = −0.02). CSF SHBG is not a suitable biomarker to discriminate FTLD-Tau from FTLD-TDP.

## 1. Introduction

Frontotemporal lobar degeneration (FTLD) covers a spectrum of highly heterogeneous disorders from a clinical, genetic and pathological perspective [1]. The majority of FTLD cases are neuropathologically characterized by either the presence of aggregates of proteins tau (FTLD-Tau) or TDP43 (FTLD-TDP), which likely requires distinct pharmacological therapy. However, the clinical presentation of these pathological subtypes is heterogeneous and overlapping, making this diagnostic subtyping at present difficult. Thus, there is a strong unmet need to identify body-fluid-based biomarkers discriminating FTLD pathological subtypes.

Most biomarker studies performed to date have analyzed pathologically heterogeneous populations [2]. The few studies analyzing antemortem cerebrospinal fluid (CSF) with known underlying neuropathology have revealed several candidate biomarkers (e.g., p/tTau ratio) [2,3,4]. However, their specificity is still not optimal and most of the identified markers are awaiting further validation [2]. Our previous unbiased proteomics study revealed lower levels of sex hormone-binding globulin (SHBG) in the CSF of FTLD-Tau cases compared to FTLD-TDP or non-demented controls [4], suggesting that SHGB could be a potential biomarker for FTLD related tauopathies. SHBG is an androgen transport protein, and thus regulates the bioavailability of sex hormones. Despite their role on FTLD remains unknown, sex hormones have been associated to cognitive function, neurodegeneration and, more specifically, to Tau proteostasis [5,6]. In this study, we aimed to validate whether CSF SHBG is a useful biomarker for discriminating the main FTLD pathological subtypes in an independent cohort using an automated electro-chemiluminescent immunoassay (ECLIA), a serum test utilised in routine diagnostics. Next, the relationship of SHBG with cognitive function and other CSF markers known to be changed in FTLD or between FTLD subtypes (i.e., YKL40 and the ratio of phosphorylated Tau at 181 to total Tau (p/tTau)) was evaluated.

## 2. Methods

### 2.1. Humans CSF Samples 

The total cohort (*n* = 85) included ante-mortem CSF samples from 52 FTLD-confirmed patients and 33 cognitive unimpaired individuals that were used as controls. FTLD cases with tau neuropathology (FTLD-Tau, *n* = 23) were selected based on autopsy (*n* = 10; including 2 cases with progressive supranuclear palsy (PSP) and 2 cases with corticobasal degeneration CBD tauopathy), and the presence of mutations in the *MAPT* gene predictive of Tau proteinopathy (*n* = 8) [1]. Diagnostic groups were enriched with patients with FTD clinical syndromes that reflect high correlation with a specific neuropathology. Thus, the FTLD-Tau group was enriched with non-autopsied patients with clinical PSP (*n* = 5), highly-associated to Tau pathology [7]. FTLD patients with an underlying TDP43 pathology (FTLD-TDP, *n* = 29) were selected based on autopsy (*n* = 13) or the presence of mutations predictive of TDP43 proteopathy (9 *C9orf72* and 3 *GRN* mutation carriers) [1]. The FTLD-TDP group was enriched with CSF from non-autopsied patients with clinical FTD and amyotrophic lateral sclerosis (FTD-ALS, *n* = 4), which is associated with underlying TDP43 pathology [1]. CSF samples were selected from the Amsterdam Dementia cohort (24 non-demented controls, 6 FTLD-Tau, 12, FTLD-TDP), the Center for Neurodegenerative Disease Research at the University of Pennsylvania (8 controls, 7 FTLD-Tau and 15 FTLD-TDP), Erasmus Medical Center (1 control, 6 FTLD-Tau and 2 FTLD-TDP) and Emory University (4 FTLD-Tau). CSF was collected by lumbar puncture, and processed and stored in agreement with the JPND BIOMARKAPD guidelines [8].

All participants underwent standard neurological and cognitive assessments and diagnosis was assigned according to consensus criteria [7,9,10]. Mini-Mental State Examination (MMSE) was used as a measure of global cognition. The control group were cognitively unimpaired individuals or individuals labeled during a multidisciplinary consensus meeting as subjective cognitive decline when they presented with subjective cognitive complaints, but objective cognitive and laboratory investigations were not abnormal. Demographic and CSF biomarkers data are summarized in Table 1. The biobanks were approved by the Institutional Ethical Review Boards of each center. Informed consent was obtained from all subjects.

### 2.2. CSF Biomarker Analysis 

CSF SHBG concentration was determined using the automated ECLIA serum SHBG test (cobas e 601, Roche, Basel, Switzerland) following manufacturer’s recommendations. We first evaluated and validated the performance of the assay for CSF analysis using CSF pools (Appendix A). Dilution-linearity showed good recovery until 1:3 dilution. Spike-recovery analysis showed optimal recovery rates. Intra- and inter-coefficient of variation (CV) were 2.3% and 8.3% respectively. As CSF samples included in this study had undergone between 1–3 freeze–thaw cycles, we additionally analyzed the effects of repeated freeze–thawing on SHBG levels. We observed that SHBG levels remained stable for up to 8 freeze–thaw cycles. Samples were measured in duplicate in two different batches. Three bridging samples were used for reference sample normalization to control for potential batch effects. 

Levels of YKL40 were available for a subset of samples (*n* = 30) from a previous study [11] (MicroVue YKL-40 ELISA, Quidel, San Diego, CA, USA). CSF pTau181 and total tau are analyzed locally as part of the routine diagnostic procedure using commercially available kits (VUmc: hTAUAg, phospho-Tau(181P); Fujirebio, Ghent, Belgium; Penn and Emory: Luminex xMAP INNO-BIA AlzBio3; Austin, TX, USA).

### 2.3. Statistical Analysis

Statistical analyses were performed in SPSS version 27 (IBM Corp., Armonk, NY, USA) and graphs were drafted with GraphPad Prism 9 (GraphPad, La Jolla, CA, USA). Between-group analyses of demographic variables were performed using the Student’s t-test or Pearson’s chi-square test after normalizing skewed data using two-step transformation. Regression and Spearman correlation analyses were performed to analyze the relationship between SHBG, age and sex. Differences in SHBG concentration between clinical groups were evaluated by general linear modeling using normalized values and including age as covariate. Diagnostic performance was evaluated using receiver operator characteristic (ROC) and areas under the curve (AUC). Partial non-parametric correlation analyses were performed to analyze the association between CSF SHBG with CSF YKL40, p/tTau ratio and MMSE score including age as covariates. Passing-Babock transformation formulas were calculated based on individuals with both Luminex and Innotest values for tTau and pTau to estimate the equivalent Innotest values for those samples measured with Luminex platform only.

## 3. Results

Analysis of demographic characteristics showed that sex did not differ between FTLD and controls. However, nominal significance indicated a higher number of females in the FTLD-Tau group and higher number of males in the FTLD-TDP group. FTLD and its pathological subtypes had lower MMSE than controls. FTLD patients had lower CSF p/tTau ratio and higher YKL40 levels compared to controls. CSF p/tTau ratio was lower in FTLD-TDP compared to both FTLD-Tau and controls (Table 1). 

No difference in CSF SHBG levels was observed between females and males (Figure 1A), neither after stratifying for diagnosis type. CSF SHBG levels were positively associated with age (r = 0.3, *p* = 0.006). Stratification into clinical groups showed that this correlation was significant in the FTLD group only (Spearman-rho = 0.35, *p* = 0.01, Figure 1B). CSF SHBG was comparable between the different diagnostic groups (Figure 1C), though a tendency towards elevated levels of CSF SHBG in FTLD-Tau compared to FTLD-TDP and controls was detected (*p* = 0.051 and 0.066, Figure 1C). Exploratory graphical evaluation within FTLD subgroups suggests that CSF SHBG was not associated to any specific subgroup within each FTLD subtype (i.e., FTD clinical syndrome, specific mutation or autopsy phenotype; Figure 1C). CSF SHBG could not discriminate FTLD-Tau from FTLD-TDP (AUC: 0.64, *p* > 0.05, Figure 1D), and did not provide any additional diagnostic value when combined with p/tTau ratio (*p* > 0.05). 

We next investigated the relationship of CSF SHBG with other FTLD CSF markers and with cognitive function. CSF SHBG did not correlate with YKL40 or the p/Tau ratio (Figure 1E–G). However, we detected an inverse correlation between CSF SHBG and MMSE (r = −0.31, *p* = 0,01; Figure 1G). Stratification into pathological subtypes suggest that such association was likely driven by the FTLD-Tau group (r = −0.38, *p* > 0.05, Figure 1G).

## 4. Discussion

This study reveals that CSF SHBG, as measured by an automated ECLIA, did not differ across pathological groups, though a trend towards increased levels of CSF SHBG was observed in FTLD-Tau cases compared to FTLD-TDP and controls. CSF SHBG levels were inversely associated with cognitive function.

The increasing trend of CSF SHBG levels in FTLD-Tau patients detected in this study may become significant when analyzing larger data sets. However, the CSF SHBG overlap between FTLD-Tau and FTLD-TDP will likely still be too high to be a useful discriminatory marker for FTLD pathological subtypes. This is further supported by the low and variable discriminative performance of SHBG detected in this study. The results of this study counters our previous unbiased proteomic findings in which CSF SHBG was decreased in FTLD-Tau patients compared to both controls and FTLD-TDP [4]. Translation of proteomic findings into high-throughput immunoassays for routine analysis is a major hurdle in the development of optimal biofluid-based biomarkers and highlights the importance of subsequent validation studies [12]. Our previous mass-spectrometry results detected SHBG peptides covering the complete protein sequence, and thus it seems unlikely that the divergent findings might be explained by the detection of different protein sequences or isoforms through the different technologies. Despite the fact that we analyzed pathologically-confirmed FTLD cases in both studies, each FTLD pathological subtype is highly heterogeneous and can cover a wide range of different pathological phenotypes [1]. Importantly, it is still not clear whether sporadic and familial cases, or even the different mutations with converging proteopathies (e.g., *GRN* or *C9orf72*), share the same pathophysiological processes, which can ultimately impact the body-fluid biochemical profiles [13,14,15,16,17]. This study included a considerable amount of autopsy-confirmed FTLD-Tau cases and several patients harboring *C9orf72* mutation or with FTLD-ALS, which were not analyzed in our previous study. There was, however, no apparent pattern associated with the specific clinic-pathologic and genetic subgroups, suggesting that within group heterogeneity may not explain the contrasting findings. The FTLD-Tau group of the previous study was younger than the patients included in this study. Considering CSF SHBG increases with age, demographic differences may partly explain the discrepancies observed. Different disease stages at time of sample collection could also contribute to the difference in findings. Analyzing the effect of the disease stage on FTLD biomarkers is challenging, especially considering there is not yet an established definition of the prodromal phase of FTLD [18]. It is also worth noting that similar inconsistencies on SHBG levels across studies analyzing blood from Alzheimer’s disease (AD) patients have also been observed [19,20,21,22]. 

Despite serum SHBG levels being often elevated in females [19,23], we did not observe such changes in CSF, which may indicate that sex related SHBG differences might be more pronounced in blood than in CSF [16]. The lack of association of CSF SHBG with CSF YKL-40 or p/tTau ratio suggests that SHBG is not related to neuroinflammatory processes or Tau proteostasis.

We observed that CSF SHBG was inversely associated, albeit weakly, with cognitive function, which is in line with previous findings showing higher levels of serum SHBG associated with risk of dementia or cognitive decline [19,24]. Interestingly, such association was likely driven by the FTLD-Tau, which showed a correlation coefficient considerably stronger than the FTLD-TDP group (−0.38 vs. −0.02). Previous studies have shown that not only SHBG but also other hormones (e.g., estrogen, free thyroxin or thyroid-stimulating hormone) are associated with MMSE [22,23], suggesting that changes in the whole hypothalamic-pituitary (HP) axes may influence memory function and the risk of dementia or AD [24,25]. This is supported by different neuropathological and imaging studies, highlighting the importance of HP dysfunction (e.g., body weight, circadian rhythm, sleep) in AD pathophysiology [25]. The association of CSF SHBG with MMSE detected in this study suggests that such processes might be also important in FTLD-Tau. Indeed, hypothalamic alterations have already been identified in the behavioral form of FTD [26], in which a more prominent hypothalamic proteopathy was observed in FTLD-Tau than in FTLD-TDP cases [27].

Despite the use of well-characterized samples from cases with confirmed FTLD diagnosis is a strength of this study, we did not have access to samples from patients with an underlying FUS pathology, who account for approximately 5% of all FTLD cases [1]. We acknowledge that the sample size to perform additional subgroup analysis within each pathological subtypes (e.g., sporadic, specific underlying mutation) is still limited. Such a type of sub-analysis requires the inclusion of large set of samples with post-mortem information and deep phenotyping, underpinning the need for high-collaborative multicenter studies and the extension and development of patients’ cohorts within the FTLD field. Yet, the sample size of pathologically-confirmed FTLD cases is still comparable or even larger to the ones used in previous pathological FTLD biomarker studies, including ours [3,4,11]. In addition, we still do not know whether the disease stage at time of collection contributes to the difference in findings on SHBG observed across studies. The incorporation of the newly-developed psychometric measures to characterize the different FTLD symptomatic stages (e.g., FTLD-CDR, MIR) may facilitate validation of FTLD biomarkers [18,28]. Studies on larger datasets are still needed, as they will allow more complex statistical modelling that can account for the heterogeneity of disease in more detail to resolve these issues.

In conclusion, our results indicate that, despite the fact that CSF SHBG tends to be increased FTLD-Tau, there is high overlap between controls and FTLD pathological subtypes. We showed that CSF SHBG is not a suitable biomarker to discriminate FTLD pathological subtypes using the automated immunoassay employed in this study, and thus, the quest for novel body-fluid based biomarkers deciphering FTLD pathological complexity will continue.

## Figures and Tables

**Figure 1 biomolecules-11-01484-f001:**
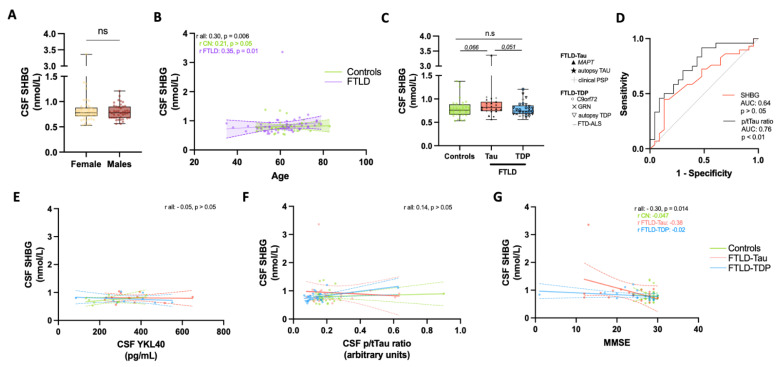
CSF SHBG levels of FTLD of patients with known underlying neuropathology. (**A**) Box-dot plots depict the CSF SHBG levels between males and females. (**B**) Spearman correlation analysis depicts an association between SHBG and age, which was mainly driven by the FTLD group (purple). Correlation coefficient and *p* value are presented in inserts. Regression line and 95% confident intervals for FTLD (purple) and controls (green) are included. (**C**) Box-dot plots depict the CSF SHBG levels between controls, FTLD-Tau and FTLD-TDP. Specific symbols depict patient subgroups within each pathological subtype. Exact *p* values after post-hoc analysis are included. (**D**) Receiver operator characteristic (ROC) curve depicting the predictive performance of CSF SHBG (red) or CSF p/tTau ratio (black) discriminating FTLD-Tau from FTLD-TDP subtypes. The exact area under the curve (AUC) and *p* value are presented in inserts. (**E**–**G**) Scatter plots depict the correlation between CSF SHBG with (**E**) CSF YKL40, (**F**) p/tTau ratio and (**G**) MMSE score. Correlation coefficient and *p* value are presented in inserts. Regression line and 95% confident intervals are included. The different diagnostic groups are depicted by colors. CN, controls; FTLD, Frontotemporal lobar degeneration; PSP, progressive supranuclear palsy; FTD-ALS, Frontotemporal dementia—Amyotrophic lateral sclerosis; ns, non-significant.

**Table 1 biomolecules-11-01484-t001:** Demographic characteristics.

	*n* (F/M)	Age (Mean, SD)	MMSE * (Mean, SD)	CSF p/tTau ^†^	CSF YKL40 ^‡^ (ng/mL)	SHBG (nmol/L)	Subgroups			
**CON**	33 (17/16)	62 (8)	28 (1)	0.19 (0.06)	234 (138)	0.8 (0.21)	na			
**FTD**	52 (26/26)	61 (10)	24 (6) ^§^	0.16 (0.08) ^§^	349 (133) ^§^	0.85 (0.21)	na			
**FTLD-Tau**	23 (16/7)	60 (13)	23 (6) ^§^	0.19 (0.07)	363 (109) ^§^	0.93 (0.20)	10 Autopsy, 8 *MAPT*, 5 cPSP			
**FTLD-TDP**	29 (10/19)	62 (6)	24 (7) ^§^	0.12 (0.06) ^§,¶^	332 (166) ^§^	0.78 (0.18)	13 Autopsy, 9 *C9orf72*, 3 *GRN*, 4 FTD-ALS			

Data are reported as medians and interquartile range unless indicated.*, ^†^, ^‡^ Data available from 132 controls and 38 FTD (18 Tau, 20 TDP) *; 227 controls and 31 FTLD (16 FTLD-Tau, 15 FTLD-TDP) ^†^; 8 controls and 22 FTLD (12 FTLD-Tau, 10 FTLD- TDP) ^‡^. F, female; M, male; MMSE, Mini mental Score Examination; CON, Control, FTLD, frontotemporal lobar degeneration; MAPT, Microtubule Associated Protein Tau mutation carriership; cPSP, clinical Progressive supranuclear palsy; C9orf72, chromosome 9 open reading frame 72 mutation carriership; GRN, progranulin mutation carriership; FTD-ALS, Frontotemporal dementia—Amyotrophic lateral sclerosis. *p* value 0.05 ^§^ vs. control or vs. ^¶^ FTLD-Tau.

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
