# Peer review of "Sex Hormone-Binding Globulin (SHBG) in Cerebrospinal Fluid Does Not Discriminate between the Main FTLD Pathological Subtypes but Correlates with Cognitive Decline in FTLD Tauopathies"

_biomolecules, 2021, doi:10.3390/biom11101484_

Round 1

Reviewer 1 Report

The authors have made an adequate effort to improve the paper by further analysis.  There is some interesting features to the paper.  The study was underpowered but it is appreciated that collecting enough samples from this more rare form of dementia is hard, especially due to the heterogeneous nature of "FTD"  I would suggest that adding another sentence to introduction concerning the biology of SHBG and neurodegeneration.  Adding another sentence extracted from ref 5, 6 would make the paper more interesting to readers.

Reviewer 2 Report

The Authors have sufficiently addressed the Reviewer's comments and queries, and have improved the quality of the manuscript.

This manuscript is a resubmission of an earlier submission. The following is a list of the peer review reports and author responses from that submission.

Round 1

Reviewer 1 Report

The Authors take further their previous study on sex hormone-binding globulin (SHBG) level in cerebrospinal fluid in frototemporal lobar degeneration. The topic is important, the method is adequate and the paper is well written. However, there are major limitations and some minor concerns, as detailed below:

1) Case numbers are rather low. The authors acknowledge this as a major limitation and the Reviewer agrees. In view of i) further subtypes in FTLD-TDP and ii) non-significantly higher level in FTLD-tau for this is particularly relevant.

2) A major FTLD type, FTLD-FUS, has not been studied.

3) Is there correlation with disease stage/severity? (time from disease onset, degree of atrophy on imaging, MMSE rate of decline, etc.)

4) The association of SHBG level with decreased MMSE is a major positive (although not new) finding. Therefore, Title change is suggested: ‘Sex hormone-binding globulin (SHBG) level in cerebrospinal fluid correlates with cognitive decline in frontotemporal lobar degeneration irrespective of neuropathological subtype’. As mentioned above, FTLD-FUS cases have to be included to test this statement.

5) Formatting suggestion: omit division of words (cere-brospinal; pa-tients) in the Title

6) Add male/female ratio or case numbers to Table 1.

Reviewer 2 Report

There are limited amounts of data that are showing significant differences between disease groups.  The topic is significant but the authors have not provided limited data to move the field forward.  The only data that saves the paper is fig 1F.  The significance level is weak but there is something there that might be exploited.  I would suggest that the authors analyze the data more intensively to determine whether the biomarker levels in different disease groups also maintain the significance with MMSE levels.  I would also like to see if combining YKL40 and SHBG measures and analyzying for MMSE would be interesting.  The other issue is whether AUC-sensitivity/specifity analysis should be made.  This should be a simple but useful addition to enhance the paper.  At present, the data in such a paper is thin.